# Urine Protein/Creatinine Ratio in Thrombotic Microangiopathies: A Simple Test to Facilitate Thrombotic Thrombocytopenic Purpura and Hemolytic and Uremic Syndrome Diagnosis

**DOI:** 10.3390/jcm11030648

**Published:** 2022-01-27

**Authors:** Laure Burguet, Benjamin Taton, Mathilde Prezelin-Reydit, Sébastien Rubin, Walter Picard, Didier Gruson, Anne Ryman, Cécile Contin-Bordes, Paul Coppo, Christian Combe, Yahsou Delmas

**Affiliations:** 1Service de Néphrologie Transplantation Dialyse Aphérèses, Centre de Référence Maladies Rénales Rares du Sud-Ouest, Centre Hospitalier Universitaire de Bordeaux, 33000 Bordeaux, France; laure.burguet@gmail.com (L.B.); benjamin.taton@chu-bordeaux.fr (B.T.); mathilde.reydit@gmail.com (M.P.-R.); sebastien.rubin@chu-bordeaux.fr (S.R.); christian.combe@ch-bordeaux.fr (C.C.); 2Service de Réanimation, Centre Hospitalier de PAU, 64000 Pau, France; walter.picard@ch-pau.fr; 3Service de Réanimation, Centre Hospitalier Universitaire de Bordeaux, 33000 Bordeaux, France; didier.gruson@chu-bordeaux.fr; 4Laboratoire d’hématologie Biologique, Centre Hospitalier Universitaire de Bordeaux, 33000 Bordeaux, France; anne.ryman@chu-bordeaux.fr; 5Département d’hématologie et Centre de Reference des Microangiopathies Thrombotiques, Hôpital Saint-Antoine, Assistance Publique des Hôpitaux de Paris and Sorbonne Universités, 75651 Paris, France; cecile.bordes@chu-bordeaux.fr (C.C.-B.); paul.coppo@aphp.fr (P.C.); 6Laboratoire d’Immunologie et Immunogénétique, Centre Hospitalier Universitaire de Bordeaux, 33000 Bordeaux, France; 7Immunoconcept, CNRS UMR 5164, Université de Bordeaux, 33076 Bordeaux, France; 8Unité INSERM 1026 Biotis, Université de Bordeaux, 33076 Bordeaux, France

**Keywords:** diagnosis, differential, hemolytic-uremic syndrome, proteinuria, purpura, thrombotic thrombocytopenic, thrombotic microangiopathies

## Abstract

Background: Early diagnosis of thrombotic thrombocytopenic purpura (TTP) versus hemolytic and uremic syndrome (HUS) is critical for the prompt initiation of specific therapies. Objective: To evaluate the diagnostic performance of the proteinuria/creatininuria ratio (PU/CU) for TTP versus HUS. Patients/Methods: In a retrospective study, in association with the “French Score” (FS) (platelets < 30 G/L and serum creatinine level < 200 µmol/L), we assessed PU/CU for the diagnosis of TTP in patients above the age of 15 with thrombotic microangiopathy (TMA). Patients with a history of kidney disease or with on-going cancer, allograft or pregnancy were excluded from the analysis. Results: Between February 2011 and April 2019, we identified 124 TMA. Fifty-six TMA patients for whom PU/CU were available, including 35 TTP and 21 HUS cases, were considered. Using receiver–operating characteristic curves (ROC), those with a threshold of 1.5 g/g for the PU/CU had a 77% sensitivity (95% CI (63, 94)) and a 90% specificity (95% CI (71, 100)) for TTP diagnosis compared with those having an 80% sensitivity (95% CI (66, 92)) and a 90% specificity (95% CI (76, 100) with a FS of 2. In comparison, a composite score, defined as a FS of 2 or a PU/CU ≤ 1.5 g/g, improved sensitivity to 99.6% (95% CI (93, 100)) for TTP diagnosis and enabled us to reclassify seven false-negative TTP patients. Conclusions: The addition of urinary PU/CU upon admission of patients with TMA is a fast and readily available test that can aid in the differential diagnosis of TTP versus HUS alongside traditional scoring.

## 1. Introduction

Our understanding of the mechanisms driving thrombotic microangiopathies (TMA) pathology has greatly improved over the past two decades. Among the various forms of TMA, thrombotic thrombocytopenic purpura (TTP) is attributed to a severe deficiency of a disintegrin and metalloprotease with thrombospondin type I repeats 13 (ADAMTS13) activity, leading to the accumulation of high-molecular-weight von Willebrand Factor multimers in plasma, resulting in excess platelet aggregation in multiple organs [1]. In contrast, hemolytic and uremic syndrome (HUS) arises as a result of endothelial dysfunction, such as that seen in Shigatoxin-producing *Escherichia coli*-associated HUS (STEC-HUS) or complement-mediated HUS (CM-HUS) [2,3].

Mortality was as high as 90% prior to the development of current treatments, but despite diagnostic and therapeutic advances, TMA remains a life-threatening disease. Recent advances in the understanding of TMA pathology and the discrimination between TMA subtypes has played a central role in the development of therapies, with eculizumab now used as first-line therapy for CM-HUS [4], compared with a combination of plasma exchange, rituximab and caplacizumab for the treatment of acquired TTP [5,6,7]. Given these divergent treatments and the pace of disease progression, the ability to differentiate TMA features rapidly is essential for providing specific life-saving treatment to these patients. Currently, ADAMTS13 activity is the only measurement able to unequivocally distinguish TTP from other TMA subtypes [8]. However, the usual turnaround time of ADAMTS13 activity, when available, is anywhere from 2 to 7 days. This major limitation in rapidly distinguishing TTP and HUS has prompted investigators to establish a scoring system based on standard clinical features available within a few hours of emergency room admission.

In 2010, our French TMA reference centergroup identified the association of two main predictive criteria for TTP diagnosis: serum creatinine level (sCr) < 200 µmol/L and platelet count < 30 G/L [9].These criteria, named the “French Score” (FS), are used in clinical practice for TTP diagnosis [6,7].

More recently, the PLASMIC score confirmed the strength of these biomarkers in patients with features of thrombotic microangiopathy with additional factors (hemolysis, mean corpuscular volume, INR and history of cancer or transplantation) [10,11].

Given that high-range proteinuria is commonly observed in HUS, and as kidney involvement is more predominant in HUS than in TTP, we hypothesized that proteinuria could represent an additional marker that could be used to improve the current diagnostic criteria. The aim of this study was to evaluate the diagnostic performances o fthe proteinuria/creatininuria ratio (PU/CU) on a urine spot sample at admission in association with the FS for the differential diagnosis of TTP versus HUS. 

## 2. Patients and Methods

### 2.1. Patients

This retrospective study included all patients > 15 years of age with TMA features, who underwent an ADAMTS 13 assay at the University Hospital of Bordeaux, between February 2011 and April 2019.

### 2.2. Definitions

ADAMTS13 activity < 10% defined TTP [12]. ADAMTS13-detectable patients werefurther classified as either STEC-HUS for patients who tested positive for Shigatoxin in stool or positive Shigatoxin-secreting *E.**coli* serology or CM-HUS in cases of confirmed or suspected abnormal regulation of alternative complement pathway [13].

We decided to include only patients with a well characterized and non-debatable diagnosis of microangiopathy by including only primary TMA (TTP, and CM-HUS) or shigatoxin documented HUS. Additionally, we excluded patients with a medical history of conditions associated with possible proteinuria (previous known kidney disease or TMA occurring in the setting of pregnancy). 

In summary, we excluded TMA occurring in the setting of pregnancy, cancer or allograft (bone marrow or solid organ) and patients with previous known kidney disease unrelated to TMA. For patients with multiple acute episodes of TMA during the inclusion period, we analyzed only one episode. 

### 2.3. Clinical and Laboratory Parameters

Final diagnosis, demographic features and organ involvement, including cardiac involvement (increased troponin level) and neurological involvement (objective neurological symptoms and ischemic lesions or thrombotic microangiopathy features in magnetic resonance imaging when available) were collected from patient medical records. Biological values included hemoglobin, platelet count, reticulocytes, LDH level, troponin level, sCR, PU/CU, and albuminuria/creatininuria ratio (AU/CU)on-admission urinary spot. FS values were calculated as described previously [9].

### 2.4. Specific Analyses

ADAMTS13 assays were performed using thefull *von Willebrand factor length* technique [14]. Patients with HUS were explored for complement proteins (CH50, C3, C4, complement factors H, I, B and MCP), anti-factor H antibodies and screened for complement gene variants at the French Reference Center (Dr. Véronique Frémeaux-Bacchi, Hôpital Européen Georges Pompidou, APHP-Paris, France).

Stool Shigatoxin PCR and serum sample STEC serology were performed locally or at the French Reference Centers for Shigatoxin-related HUS (Hôpital Robert Debré and Institut Pasteur, Paris, France). 

### 2.5. Statistical Analysis

Summary statistics are presented as the median (1st quartile, 3rd quartile), or number (percentage). Intergroup comparisons were performed using bilateral Wilcoxon exact tests or χ^2^ exact tests. The statistical significance threshold was chosen to be 0.05.

First, we investigated separately the diagnosis performances of sCR, platelet count, and PU/CU for the diagnosis of TTP versus HUS by analyzing receiver operating characteristic curves (ROC). Decision thresholds were determined so as to maximize the Youden’s index.

We then compared the sensitivities, specificities, PPVs, NPVs, and error rates of the following dichotomous criteria:FS = 2, namely: platelet count < 30 G/L and Scr < 200 µmol/L,The combined criterion of FS = 2 or PU/CU ≤ T, where T denotes the PU/CU optimal threshold found in the ROC analysis.

The incremental value of PU/CU for the diagnosis of TTP was also assessed using the net reclassification improvement (NRI) [15]. Internal validation was performed by bootstrapping (500 resampling) [16,17,18]. As the combined criterion was expected to be more sensitive and less specific for TTP, comparisontestswere unilateral, when appropriate, or bilateral when no specific direction was expected. All the statistical analyses were conducted using the R language v. 4.0.2 [19].

### 2.6. Ethical Considerations

All data were anonymized and protected. The study was approved by the ethics committee, Groupe Publication du Comité d’Ethique du Centre Hospitalier Universitaire de Bordeaux, number CE-GP-2020-18.

## 3. Results and Discussion

### 3.1. Study Group

The study flowchart is detailed in Figure 1. From February 2011 to April 2019, among the dosages of ADAMT13 activity; we identified 124 patients with features of TMA, for whom data were available at Bordeaux University Hospital.

From this set of patients, 68 patients were excluded because of TMA occurring in the setting of pregnancy, cancer or allograft (bone marrow or solid organ), previous known kidney disease unrelated to TMA or unavailable PU/CU data, leaving 56 patients for analysis. The final study cohort included 34 patients with immune TTP, one patient with congenital TTP, 13 patients with STEC-HUS and 8 patients with CM-HUS. 

### 3.2. Characteristics of Patients on Admission

Our cohort included 37 women (66%) and 19 men, with a median age of 42.5 years. The median PU/CU was 1.41 g/g (0.63, 3.54), with an AU/CU(N = 27) of 0.40 g/g (0.16, 1.13). TTP patients exhibited a higher frequency of neurological symptoms 66% versus 29% (*p* = 0.01), more profound thrombocytopenia (11 G/L (8, 16.5), *p* = 4.8 × 10^−8^), higher reticulocyte count (106 (77.5, 165.25), *p* = 8 × 10^−5^), and higher troponin level (3.5 × upper normal limit (UNL) (1, 12), *p* = 2.5 × 10^−3^) than HUS patients (respective values of platelets 46 G/L, reticulocyte count 63 G/L, troponin 1× ULN). Hemoglobin and LDH levels were similar in both groups. HUS patients had significantly higher sCr (229 μmol/L (162, 307), *p* = 2.3 × 10^−7^), PU/CU (3.96 g/g (1.77, 5.79), *p* = 2.3 × 10^−7^) and AU/CU (1.95 g/g (0.77, 3.43), *p* = 2.2 × 10^−3^) (Figure 2).

### 3.3. Individual Diagnostic Powers of Serum Creatinine Level, Platelet Counts and PU/CU

We selected three variables to differentiate TTP from HUS at the time of admission: sCr, platelet count and PU/CU. Average AUC values for each parameter were 0.89 (95% CI (0.78, 0.97)) for sCr, 0.90 (95% CI (0.81, 0.97)) for platelet count and 0.89 (95% CI (0.79, 0.96)) for PU/CU. In this cohort, the optimal thresholds to distinguish between TTP and HUS were143 µmol/L(95% CI (126, 84)) for sCr, 20 G/L (95% CI (11.5, 39.5)) for platelet count, and 1.5 g/g (95% CI (1.15, 3.40)) for the PU/CU. The threshold of 1.5 g/g for the PU/CU was associated with a 77% sensitivity (95% CI (63, 94)) and a 90% specificity (95% CI (71, 100)) for TTP diagnosis.

### 3.4. Performance of a Standard French Scoreof2

In this cohort, a FS of 2 (sCr < 200 µmol/L and platelet count < 30 G/L) exhibited an 80% sensitivity (95% CI (66, 92)), a 90% specificity (95% CI (76, 100)), a 93.3% PPV (95% CI (83, 100)) and 73% NPV (95% CI (55, 89)) for TTP diagnosis. Using these criteria, nine patients were misclassified, including two false positives and seven false negatives for TTP diagnosis (Figure 3).

### 3.5. Extending a French Scoreof Twoto Account for PU/CU

A diagnostic criterion defined as FS = 2 or PU/CU ≤ 1.5 g/g performed better than the criterion FS = 2 alone in detecting TTP. Using this composite criterion, no patient in our cohort was classified as false negative for TTP, and only four were false positives for TTP diagnosis (two STEC-HUS, and two CM-HUS with MCP mutations).These patients had a PU/CU > 1.5 g/g, and a FS = 2.

With the composite criterion, sensitivity increased to 99.6% (95% CI (93, 100), *p* = 8 × 10^−3^), and NPV increased to 99% (95% CI (88, 100), *p* = 0.01). As a counterpart, the specificity of the test for the diagnosis of TTP decreased to 77% (95% CI (56, 97), *p* = 0.005). Similarly, PPV decreased to 88% (95% CI (77, 99)); however, this difference was not statistically significant (*p* = 0.08).

Overall, the optimism-corrected NRI did not differ significantly from 0 (NRI: 0.059, 95% CI (−0.18, 0.28), *p* = 0.63), but its event component did, resulting in 19% (N = 7) of TTP patients being correctly reclassified (95% CI (6, 33), *p* = 6 × 10^−3^).Similarly, the non-event component was also not significantly different from 0, with 13% of HUS patients inappropriately reclassified (95% CI (1.5, 35), *p* = 0.1). The seven patients classified for TTP had only one of the two main FS criteria and included two patients with severe kidney involvement, with a sCr ≥ 200 µmol/L, and five patients with a platelet count ≥ 30G/L. However, all of these patients exhibited a PU/CU ≤ 1.5 g/g.

Acquired TTP is associated with little or no kidney involvement [9,20]. In contrast, kidney involvement is much more frequently observed in constitutive TTP, with some patients progressing to end stage renal disease. While the mechanisms underlying this difference are not well understood [21,22], patients with other causes of kidney involvement or pregnancy were excluded from the analysis. Virtually almost all of our cases of constitutive TTP were observed in the context of pregnancy and associated with preeclampsia and were, thus, excluded from analysis.

Kidney involvement is frequently observed in HUS patients and is associated with severe glomerular lesions. Among pediatric STEC-HUS patients, Didailler et al. described a median sCr of 294 µmol/L and a median for PU/CU of 9.2 g/g [23]. Deregulation of the alternative complement pathway is also associated with distinct kidney lesions similar to those seen in glomerular TMA, arteriolar TMA and cortical necrosis [24,25]. These observations are consistent with the data presented here, in which a PU/CU ≤ 1.5 g/g alone was sensitive enough to exclude HUS diagnosis with a 90% specificity (95% CI (71, 100)).

In this study, a direct comparison with the PLASMIC score was not possible, as the data required to calculate the score were not collected in the entire cohort [10]. In the context of TMA, clinicians’ priority remains the detection of TTP, so any improvements in diagnostic score would prioritize sensitivity over specificity.

Eight TTP patients presented with PU/CU > 1.5 g/g, without other signs of kidney involvement (sCr < 200 µmol/L). In some cases, the high rate of PU/CUin TTP patients may have been due to hemoglobinuria, meaning AU/CU could be a useful tool for differentiating between specific glomerular involvement and hemolysis-inducedtubulopathy.

In recent TTP studies conducted by the French TMA reference center, a FS of twowith TMA features was used to initiate immediate TTP-specific treatment with plasma exchange and caplacizumab, before ADAMTS13 confirmation results [5,6,7]. If the reduced rate of TTP misdiagnosis by PU/CU was confirmed by prospective studies, TTP-specific treatment could be proposed to patients with FS = 2 or FS = 1 with a PU/CU ratio ≤ 1.5 g/g (Figure 4).

Our study had some limitations. First of all, it was a retrospective, single-center study witha limited number of patients with available PU/CU data. These results need to be confirmed in a prospective study with a larger number of patients. Additionally, it would be interesting to evaluate the use of PU/CU in association with the PLASMIC score. Secondly, we focused on a very narrow subset of TMA patients and excluded all secondary TMA in the setting of pregnancy, cancer or allograft, despite these cases accounting for the majority of TMA diagnoses [26].

## 4. Conclusions

In patients with suspected TMA, urinary PU/CU on admission ≤ 1.5 g/g and/or a FS = 2 is a quick and simple criterion by which to suggest TTP rather than HUS, enabling the rapid initiation of targeted therapies. While suggestive, these results need to be confirmed in a prospective study involving a larger cohort.

## Figures and Tables

**Figure 1 jcm-11-00648-f001:**
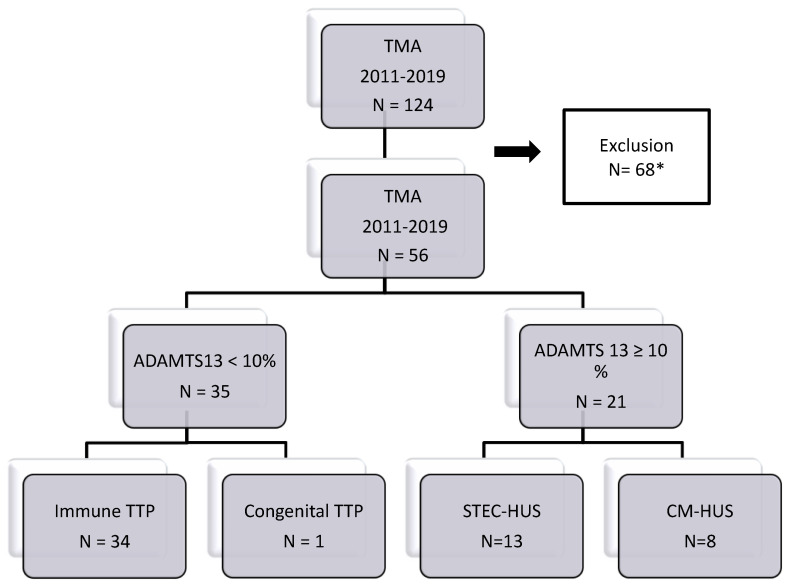
Flow-chart of the study: 56 patients were enrolled between February 2011 and April 2019. * Exclusion criteria: all TMA occurring in the setting of pregnancy, cancer or allograft, and all patients with previous kidney disease not related to TMA, and unavailable PU/CU data.

**Figure 2 jcm-11-00648-f002:**
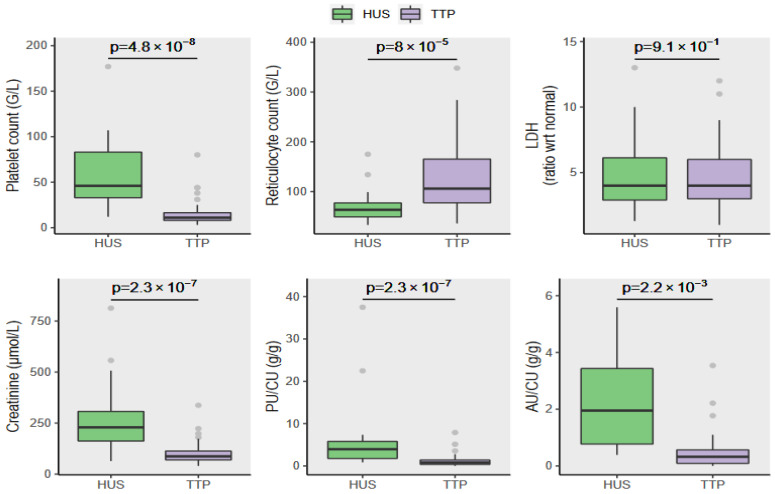
Univariate comparison of biological criteria between HUS patients (green box plots) and TTP patients (purple box plots). TTP patients exhibited significantly higher levels of thrombocytopenia and higher reticulocyte counts compared to HUS patients. LDH levels were similar in both groups. HUS patients had significantly higher serum creatinine level, PU/CU, and AU/CU.

**Figure 3 jcm-11-00648-f003:**
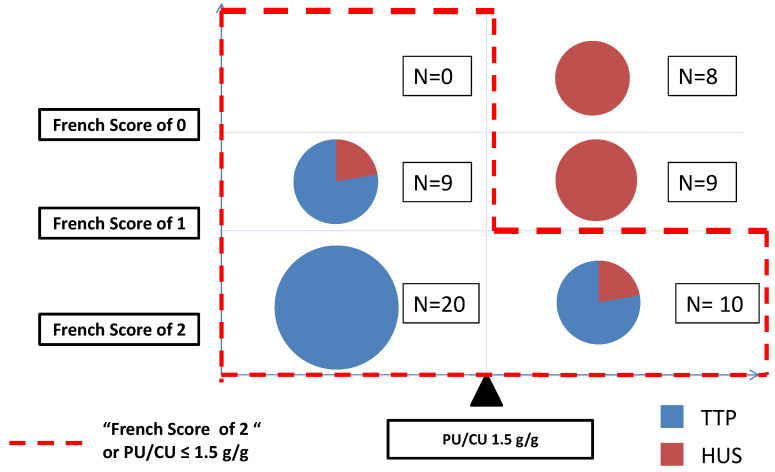
Distribution of TTP (blue), and HUS (red) diagnoses in our cohort according to French Score and PU/CU. Improved TTP diagnosis using our composite score French Score of 2 or PU/CU ≤ 1.5 g/g (red dotted line), despite four false positive results (two CM-HUS with MCP mutations, and two STEC-HUS).

**Figure 4 jcm-11-00648-f004:**
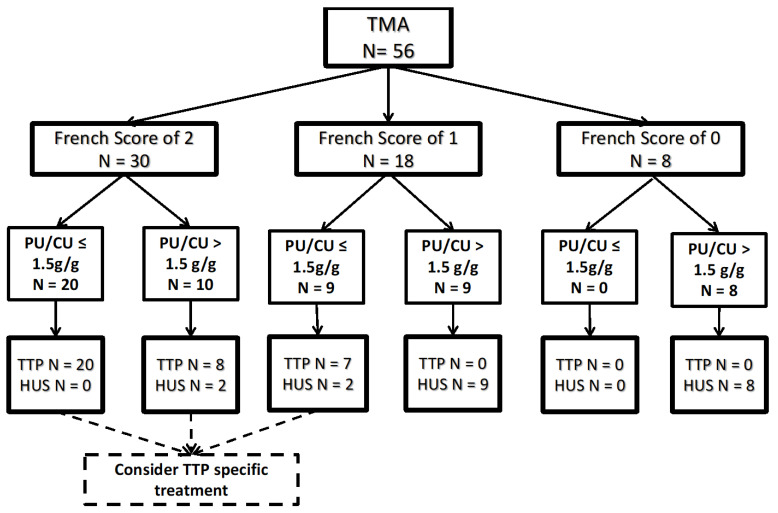
Proposed TMA diagnosis approach using the French Score and PU/CU. Improved TTP diagnosis could allow for more widespread use of specific treatments in patients with French score of 2 or French score of 1 with a PU/CU ≤ 1.5 g/g.

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
