# Peer review of "Urine Protein/Creatinine Ratio in Thrombotic Microangiopathies: A Simple Test to Facilitate Thrombotic Thrombocytopenic Purpura and Hemolytic and Uremic Syndrome Diagnosis"

_jcm, 2022, doi:10.3390/jcm11030648_

Round 1
Reviewer 1 Report
The authors showed that urine protein/creatinine ratio is a useful biomarker for quick diagnosis discriminating TTP and HUS in the present study. Although this study is a retrospective study performed in a single center, the obtained data may be useful in the clinical setting when it will be validated in a prospective and multi-center study.
Major comments
- The authors excluded the analysis of patients with secondary TTP in the present study. Please describe the reason for that.
- What is the results when the data of patients with secondary TTP are included?
Minor comment
The authors should refine the presentation style.
Author Response
Thank you for your reading and comments
- The authors excluded the analysis of patients with secondary TTP in the present study. Please describe the reason for that.
We decided to exclude secondary TMA in our analysis in order to assess the urinary protein/creatinine ratio in well-defined and homogenous TMA groups.
The well characterized diagnosis of TMA concerned primary TMA (TTP and complement mediated HUS) or well distinguishable TMA such as shigatoxin associated HUS. We selected TTP with the criteria of ADAMTS13 < 10% and well documented complement mediated HUS or shigatoxin documented HUS.
We decided to exclude pregnant patients as pregnancy may be associated with proteinuria independently from TMA. Also, “TMA-like” presentations in pregnancy may lead to false TMA diagnoses. Including these patients would have represented a possible confounding variable in our analysis; for instance, massive post-partum bleeding can present with TMA features but are not genuine TMA in the pathology definition (no kidney TMA histologic lesions and just hemostasis disorders associated with acute tubular necrosis).
We added in the manuscript the following sentence: we decided to include only patients with a well characterized and non-debatable diagnosis of microangiopathy by including only primary TMA (TTP, and CM-HUS) or shigatoxin documented HUS. Also, we excluded patients with a medical history of conditions associated with possible proteinuria (kidney disease or TMA occurring in the setting of pregnancy).
- What is the results when the data of patients with secondary TTP are included?
We included in our analysisTTP associated with infection or drug-induced. We did not include secondary TMA such as chemotherapy HUS or bone marrow transplant associated TMA for the reasons explained in the first reviewer comment. Further studies are mandatory to address the question of the urinary protein ratio utility in secondary TMA.

Reviewer 2 Report
Authors retrospectively analyzed their experience of distinguishing TTP from HUS, with addition of a simple urine examination and had a favorable result.
- Authors has stated that their result and suggestion has to be justified in a prospective way, and include more unconfirmed patients.
- Authors has combined Result and Discussion together in one section. This might save some space and time to read. However, some issues might also skip the inspection by readers. For example, authors examined the chance of false (+) and false (-) of FS by considering PU/CU. I admire this very much. But, what will be like if you do the other way round? What is the individual power and extent of false (+)(-) of the four parameters (FS、PU/CU、sCr、PLT count)?
- There are a few typos. Please find them correct. For example, Abstract, "...PU/CU, hada 77% sensitivity..." -- should be "...had a...". 2.1.Patients "...who at underwent..." -- why the "at" is here for? At the paragraph before Fig.4, "...In recent TTP studies conducted bythe French TMA reference center,..." -- should be :"by the".
Author Response
Thank you for your reading and comments
- Authors has stated that their result and suggestion has to be justified in a prospective way, and include more unconfirmed patients.
- Authors has combined Result and Discussion together in one section. This might save some space and time to read. However, some issues might also skip the inspection by readers. For example, authors examined the chance of false (+) and false (-) of FS by considering PU/CU. I admire this very much. But, what will be like if you do the other way round? What is the individual power and extent of false (+)(-) of the four parameters (FS、PU/CU、sCr、PLT count)?
These results below do not appear in the manuscripts for reasons of simplification and understanding
Platelet ≤ 30 G/L alone is a sensitive and specific test for TTP diagnosis (an 85% sensitivity, a 76% specificity, an 85% PPV and a 76% NPV), with the same rate of false negative (N=5) and false positive TTP patients (N=5). Creatinine level ≤ 200 µmol/L is a more sensitive test with a less specificity (a 94% sensitivity, a 52% specificity, a 78.5% PPV and an 85% NPV for TTP diagnosis), and with a high number of false positive TTP patients (N=9, versus 2 false negative TTP).
A FS of 2 (sCr ≤ 200 µmol/L and platelet count ≤ 30 G/L) improves specificity of the test, with less sensitivity (an 80% sensitivity, a 90% specificity, a 93.3% PPV and a 73% NPV for TTP diagnosis), and a higher number of false negatives TTP patients (N=7, versus 2 false positives TTP patients).
A PU/CU ≤ 1.5 g/g is also a specific test for TTP diagnosis, with less sensitivity (a 77% sensitivity, a 90% specificity, a 93% PPV and a 70% NPV for TTP diagnosis), and with a higher number of false negatives TTP patients (N=8, versus 2 false positive TTP).
With the composite criterion defined as FS=2 or PU/CU ≤ 1.5 g/g, sensitivity increased to 99.6%, and NPV increased to 99%. As a counterpart, the specificity of the test for the diagnosis of TTP decreased to 77%. Similarly, PPV decreased to 88%. Using this composite criterion, no patient in our cohort was classified as false negative for TTP, and only four were false positives for TTP diagnosis (two STEC-HUS, and two CM-HUS with MCP mutations).
|
|
Sensitivity |
Specificity |
PPV |
NPV |
|
Platelet count ≤ 30 G/L |
85% |
76% |
85% |
76% |
|
Serum creatinine ≤ 200 µmol/L |
94% |
52% |
78.5% |
85% |
|
FS = 2 |
80% |
90% |
93.3% |
73% |
|
PU/CU ≤ 1.5 g/g |
77% |
90% |
93% |
70% |
|
FS=2 or PU/CU ≤ 1.5 g/g, |
99.6% |
77% |
88% |
99% |
- There are a few typos. Please find them correct. For example, Abstract, "...PU/CU, hada 77% sensitivity..." -- should be "...had a...". 2.1.Patients "...who at underwent..." -- why the "at" is here for? At the paragraph before Fig.4, "...In recent TTP studies conducted bythe French TMA reference center,..." -- shouldbe :"by the".
We apologize for the grammatical errors and commit to correct them in the attached manuscript. Our manuscript has since been reviewed by a native speaker.
